mechanical engineering

conical pick, rock, self-rotation, impact dynamics, wear

**Author for correspondence:**
Xiaohui Liu
e-mail: chd160039@chd.edu.cn

# Effect of contact characteristics on the self-rotation performance of conical picks based on impact dynamics modelling

## Xiaohui Liu[1,2] and Qi Geng[1]

[1]Key Laboratory of Road Construction Technology and Equipment of MOE, School of Construction Machinery, Chang'an University, Xi'an 710064, People's Republic of China
[2]Post-Doctoral Research Center, Sinomach Changlin Company Limited, Changzhou 213136, People's Republic of China

XL, 0000-0003-4951-9041

Conical picks have a harsh working environment and 70% of their failures are caused by wear. It has been found that conical picks can rotate while interacting with the working material. This self-rotation ability is an important factor affecting pick wear, but the self-rotation mechanism remains unclear. Researching the mechanism involves the impact and friction dynamics of the interaction between a pick and its holder. Thus, a dynamic finite-element numerical model for the pick–holder interaction is established. How the handle–holder gap width, pick body and handle lengths and pick handle diameter affect the settling time and the percentage of time with no contact is studied to compare the numerical results with the equivalent effects on the rotation angle in experiments. Doing so indicates that the main factor in the self-rotation ability of a conical pick is not the magnitude of the contact load but its duration. The present research provides a new method for revealing the mechanism for the self-rotation of conical picks.

# 1. Introduction

Conical picks are found in many excavator machines, such as on the milling drum of a shearer and the cutting head of a roadheader [1,2], as shown in figure 1. The early research into picks comprised simulations and experiments to determine their cutting performance, such as (i) their cutting load [3,4], (ii) whether they produced lumps or dust [5–7], and (iii) their specific energy consumption [8–10]. In numerical studies of picks, the main methods are the finite-element method (FEM)

**Figure 1.** Conical picks on (*a*) a shearer and (*b*) a roadheader; (*c*) self-rotation principle.

**Figure 2.** (*a*) Curve of contact load; (*b*) pick in contact with its holder and (*c*) pick not in contact with its holder.

[11–13] and the discrete-element method [14–16]. In pick experiments, different testbeds have been established, such as the coal-rock cutting testbed [17,18] and automated rotary coal/rock cutting simulator (ARCCS) testbed [19–21] used for rotary cutting, the single-pick testbed [22,23], the full-scale cutting testbed [24,25] and some modified test devices [26,27]. Research attention is now turning to pick failure. Because most pick failures are due to wear, pick wear has been investigated experimentally [26–30]. To reveal the mechanism for pick wear, the interaction between a pick and the rock material has been studied [31–33]. Now, the search is on for effective ways to reduce pick wear, with picks being made from various new materials [34] and water jets being used to assist cutting [35,36].

It has also been found that conical picks can rotate while they interact with the working material. This self-rotation ability, as it is known, involves the pick rotating relative to its holder during the interaction with the working material. This in turn leads to uniform wear around the pick, which helps to keep the pick sharp. The principle is shown in figure 1*c*. Pick self-rotation has been studied experimentally [37] and is related to both the magnitude and duration of the contact load between the pick and its holder. However, it remains unclear whether it is the magnitude or the duration of the contact load that affects pick self-rotation the most.

Accordingly, a theoretical model was established, and the contact load due to the pick–holder interaction in the cutting process was analysed for different values of the pick structural parameters. Also, the pick rotation angle was investigated experimentally for different values of the pick structural parameters, but the experimental results were not consistent with the theoretical ones. From this, it was concluded that the magnitude of the contact load is not the main factor for pick self-rotation, but rather it is the duration of the contact load. Upon receiving the cutting load, the pick comes into contact with its holder and rotates, and the resulting graph of the contact load versus time in the settling phase is assumed to resemble that shown in figure 2*a*. The pick collides with its holder and

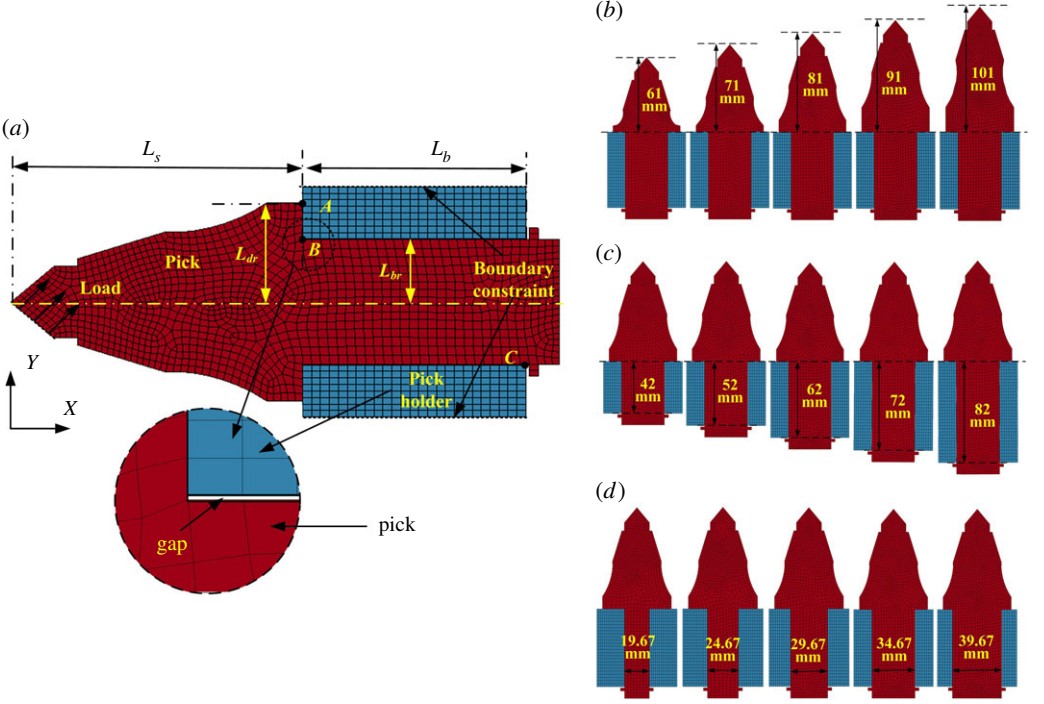

**Figure 3.** Dynamic numerical model with different values of pick structural parameters.

then loses contact with it upon rebound; this process repeats until stability is achieved and the pick is in continuous contact with its holder. In this process, the contact load trended to be stable and dense. Moreover, there are two types of time interval, namely (i) those labelled '$t_i$' in which the pick is loaded and is in contact with its holder, as shown in figure 2b, and (ii) those labelled '$t_{nj}$' in which the pick is not loaded and is not in contact with its holder, as shown in figure 2c.

The aims of the present study are to (i) verify that the main factor in pick self-rotation is the duration of the contact load rather than its magnitude, and (ii) clarify the mechanism for pick self-rotation. Studying that mechanism involves considering the pick–holder impact and the friction dynamics of the pick–holder interaction. It would be difficult to conduct experiments in which sensors were used to measure the pick dynamics during cutting (e.g. the contact load between the pick and its holder), but FEM is effective for doing so. Thus, a dynamic FEM model of the pick–holder interaction is established and used to study how the pick structural parameters affect the settling time (ST) and the percentage of time with no contact. The FEM results are compared with experimental results regarding the pick rotation angle.

## 2. Methodology

### 2.1. Numerical model

From the analysis in §1, to verify that the main factor in pick self-rotation is the duration of the contact load rather than its magnitude, the parameters related to '$t_{nj}$' must be evaluated for comparison with experimental results. This is done using a simplified two-dimensional FEM model of the pick–holder interaction as shown in figure 3a, which was established using the LS-DYNA software. The plane profile of the pick is that of an actual pick, while the pick holder is simplified as two rectangles with a gap between the pick handle and holder. Shell163 and the international system of units (m–kg–s) are applied. The pick and its holder are formed from a linear elastic material with relatively simple material parameters, namely density ($7.8 \times 10^3$ kg m$^{-3}$), elastic modulus (270 GPa) and Poisson's ratio (0.3). The cutting load acts on the carbide tip of the pick, and the boundaries of the pick holder are subject to fixed constraints. The contact type between the pick and its holder is 2D_AUTOMATIC_SINGLE_SURFACE.

The handle–holder gap width is selected as 0.05, 0.15, 0.25, 0.35 and 0.45 mm; the pick body length is selected as 61, 71, 81, 91 and 101 mm (figure 3b); the pick handle length is selected as 42, 52, 62, 72 and 82 mm (figure 3c), and the pick handle diameter is selected as 20, 25, 30, 35 and 40 mm (figure 3d). To

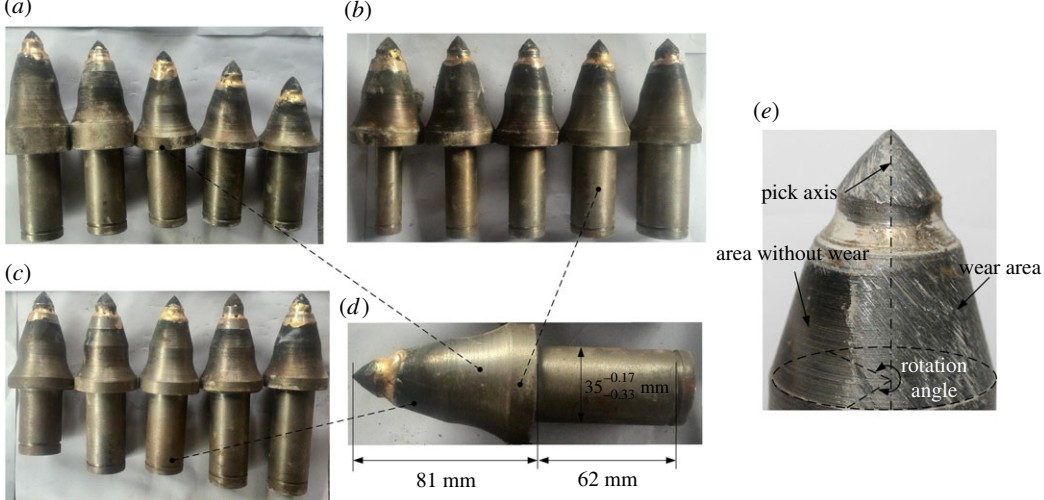

**Figure 4.** (*a–e*) Experimental picks.

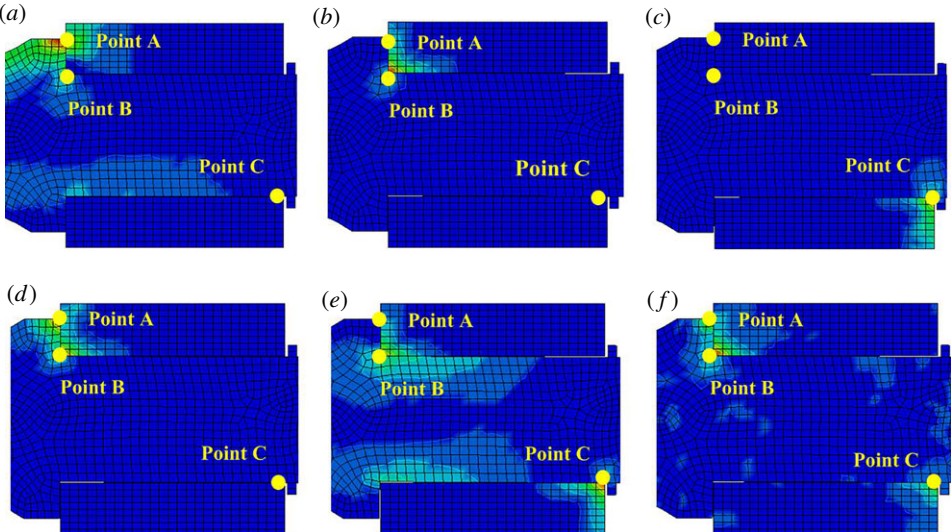

**Figure 5.** (*a–f*) Stress nephograms of pick interacting with its holder.

study the effect of the gap width, five additional models were established with diameters of 34.1, 34.3, 34.5, 34.7 and 34.9 mm.

## 2.2. Experiments

The cutting testbed is described in detail in reference [37] and is not mentioned again herein. Figure 4 shows the picks used in the experiments; the normal pick body is 81 mm in length, and the pick handle is 62 mm in length and $35.0_{-0.33}^{-0.27}$ mm in diameter. As well as the normal pick, another 12 picks were designed to study how the pick body and handle lengths and the pick handle diameter affect the pick self-rotation. Another five picks were designed to study the effect of the handle–holder gap width. All the experimental picks are the same size as those in the numerical models. In the experiments, the pick self-rotation was represented by the rotation angle (the angle of the wear area around the pick), as shown in figure 4*e*.

## 3. Results and discussion

The obtained stress nephograms for the pick interacting with its holder are shown in figure 5. These show the stress distribution in the pick and its holder under interaction. Here, red and blue correspond to the maximum and minimum loading, respectively.

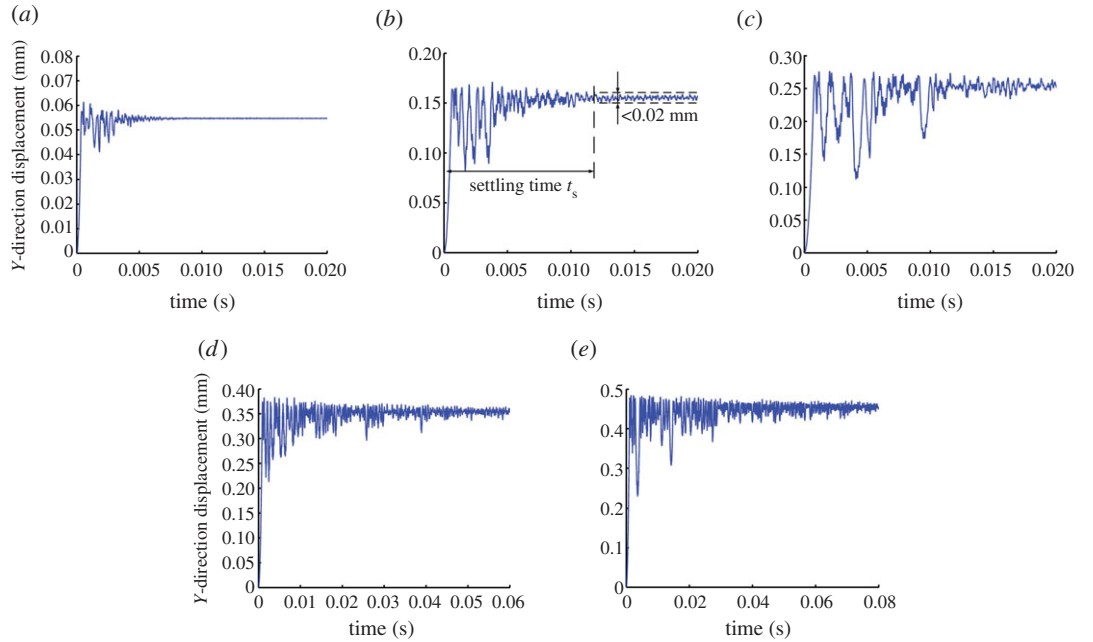

**Figure 6.** (*a*–*e*) *Y*-direction displacement of point B on pick for different gap widths.

**Table 1.** Pick parameter values used to study effect of gap width (units: mm).

| pick body length | pick handle length | pick handle diameter | inner diameter of pick holder | gap width |
|---|---|---|---|---|
| 81 | 62 | 34.9, 34.7, 34.5, 34.3, 34.1 | 35 | 0.05, 0.15, 0.25, 0.35, 0.45 |

Figure 5 shows that the pick and its holder collide and rub together all the time. They are mainly loaded at points A, B and C, but not always simultaneously. For example, in figure 5*a*–*c*, the pick and its holder are loaded at only one of points A, B and C for a time; in figure 5*d*,*e*, they are loaded at both points A and B or points B and C for another time; in figure 5*f*, they are loaded at points A, B and C simultaneously.

When the pick collides with its holder at points A and B, they lose contact because of the rebound reaction of the pick holder. They then collide at point C, followed by another rebound. The pick and its holder collide and lose contact repeatedly until they achieve stability and are in contact continuously.

## 3.1. Effect of gap width

Table 1 lists the pick parameter values used to study the effects of the gap width, and figure 6 shows the *Y*-direction displacement of point B on the pick for the five different gap width values that were tested. The simulation time is 0.02 s for the gap widths of 0.05, 0.15 and 0.25 mm. However, with the increase in the ST, the simulation time is 0.06 and 0.08 s for the gap widths of 0.35 and 0.45 mm, respectively.

### 3.1.1. Settling time

Figure 6 shows a particular ST for the *Y*-direction displacement of point B on the pick for a given gap width. Before the pick becomes steady, its displacement fluctuates because of the load on the pick and the rebound reaction of the holder. The fluctuation is large initially and then deceases with time. Once stabilized, the *Y*-direction displacement is almost the same as the gap width, and the fluctuation is much smaller than that initially.

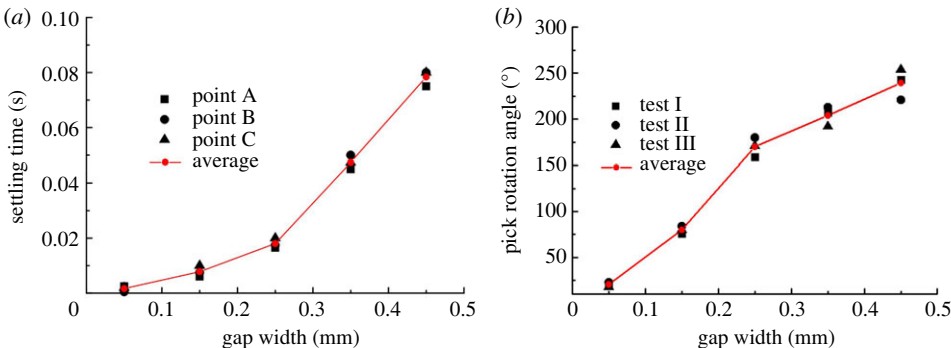

**Figure 7.** (*a*) Settling time (ST) and (*b*) rotation angle for different values of gap width.

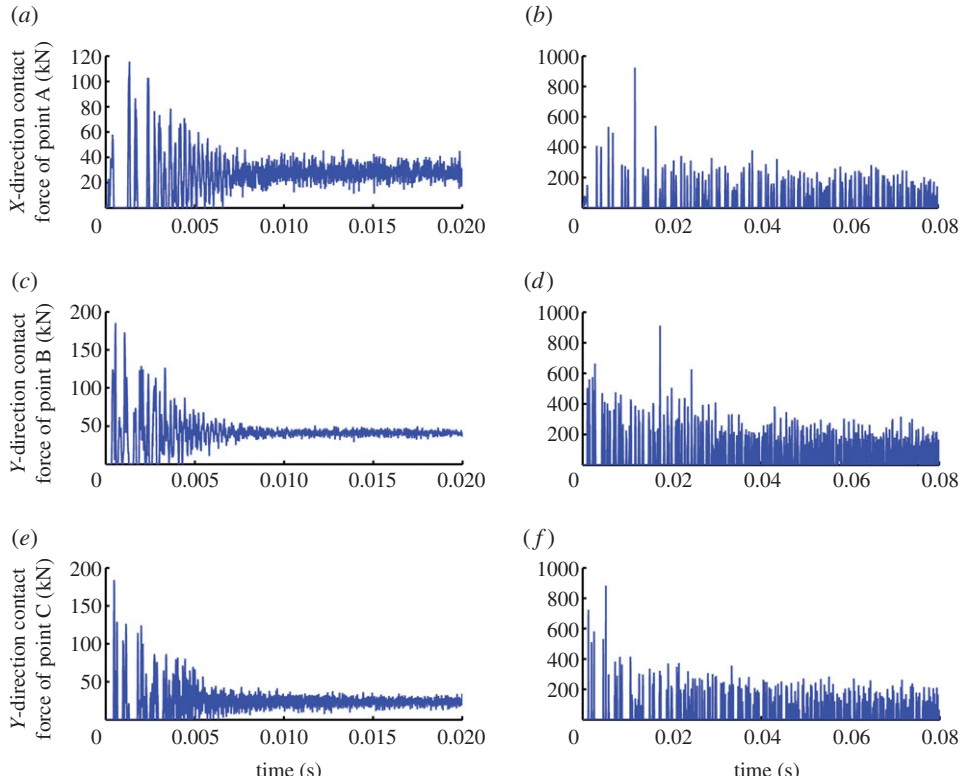

**Figure 8.** Contact forces of points A, B and C on pick for gap widths of (*a,c,e*) 0.05 mm and (*b,d,f*) 0.45 mm.

To study the stability of points A, B and C on the pick, the *X*-direction displacement of point A and the *Y*-direction displacements of points B and C are assessed simultaneously. Taking a displacement fluctuation of less than 0.02 mm as indicating that the pick is becoming steady, figure 7*a* shows the ST of each of the three points on the pick for different gap widths.

Figure 7*a* shows that the pick ST increases with the gap width, which means that the pick spends longer with no contact. This finding agrees well with the experimental results in figure 7*b*. More specifically, the implication is that the pick spends longer under little or no contact load, which favours the self-rotation of the pick in its holder. Although the gap width is necessarily limited, the STs plotted in figure 7 suggest that the self-rotation differs greatly within a gap width of 1 mm.

### 3.1.2. Contact load

Although the ST offers an index of pick self-rotation ability, it is not an accurate reflection of the length of time that the pick is subjected to a contact load. To determine that duration, the *X*-direction contact force of point A and the *Y*-direction contact forces of points B and C should be assessed. Figure 8 shows the

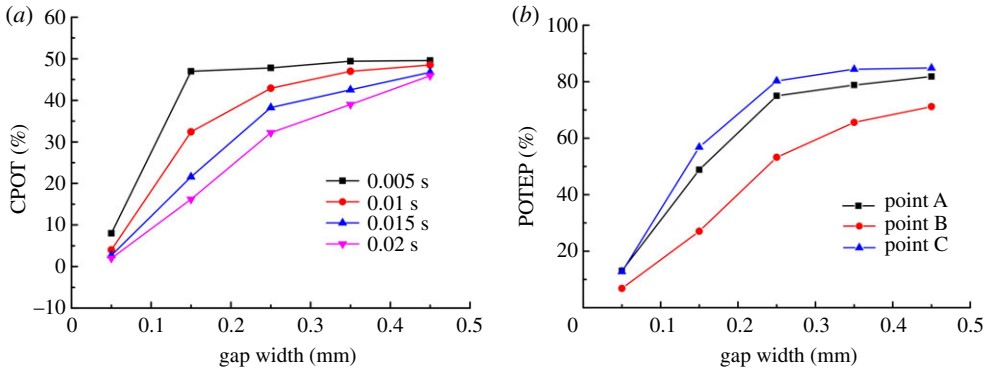

**Figure 9.** Effect of gap width on (*a*) comprehensive percentage of time with no contact (CPOT) and (*b*) percentage of time with no contact on each point (POTEP).

contact forces of points A, B and C on the pick for gap widths of 0.05 and 0.45 mm. To facilitate the analysis, the load is taken as being always positive in value.

Figure 8 shows that for the gap width of 0.45 mm, the contact forces of points A, B and C on the pick are repeatedly increasing rapidly from zero to a maximum and then decreasing rapidly to zero all the time (0.08 s). There appears a long duration that the contact force is zero. It indicates that the pick repeatedly crashed into and extruded against the holder during 0.08 s. In this case, the variation trends and the values of the contact forces of points A, B and C are almost the same as each other. On one hand, the variation trend exhibits two noticeable features: (i) the contact force is relatively large initially and then decreases to a smaller average value with time, and (ii) the contact force becomes intensive at the same time when steady, which indicates that the rebound distance of the pick has decreased and the contact frequency has increased. On the other hand, the peak forces of points A, B and C initially and after stability are the same and are approximately 400 and 200 kN. This shows that the changes in the contact force are small when the pick collides repeatedly with its holder.

For the gap width of 0.05 mm, it is only initially that the contact forces of points A, B and C on the pick increase rapidly from zero to a maximum and then decrease rapidly to zero. As time increases, the contact force fluctuates less and eventually stabilizes to a constant value, after which the contact force is never zero for a prolonged time. The results show the pick and its holder repeatedly colliding and rubbing together during the initial 0.0075 s, after which the pick achieves stability and remains in contact with its holder continuously. In this case, the variation trends of points A, B and C are almost the same as each other, although there are some differences in the values of the contact forces. On one hand, the contact force and its fluctuation are relatively large initially but then decrease and stabilize with time. On the other hand, the peak forces of points A, B and C initially are all around 120 kN, which shows again that the changes in the contact force are small when the pick collides repeatedly with its holder. However, once stable, the contact force of point A fluctuates around 30 kN, that of point B fluctuates around 40 kN, and that of point C fluctuates around 25 kN.

From the above analysis, the contact load of the pick within 0.02 s can be calculated using the contact forces of points A, B and C according to Eq. (7) in reference [37]. The percentage of sampling times at which the contact load is zero is referred to as the comprehensive percentage of time with no contact (CPOT), another index of the pick self-rotation ability. Figure 9*a* shows how the gap width affects CPOT. Moreover, to analyse how the percentage of time with no contact differs among points A, B and C, the percentage of sampling times at which the contact force is zero on each point is taken as a statistic and is defined as the percentage of time with no contact on each point (POTEP). Figure 9*b* shows how the gap width affects POTEP.

With increasing gap width, figure 9*a* shows that the contact between the pick and its holder decreases gradually: CPOT increases gradually in different time periods, but the trends are different. CPOT for the gap width of 0.05 mm differs greatly from those for the other gap widths: for the gap width of 0.05 mm, CPOT is only 8% at 0.005 s, whereas it is around 50% for the other gap widths. This indicates that the pick stabilizes in an extremely brief period and then remains in contact with its holder continuously. This is related closely to the phenomenon whereby the contact force becomes not zero at 0.0075 s. In the time period of 0.01–0.02 s, with increasing gap width, CPOT increases rapidly initially and then slowly, approaching 40–50% gradually. This means that the difference in CPOT in different time periods decreases gradually with increasing gap width. The main reason for this is that with a large gap

R. Soc. Open Sci. **7**: 200362

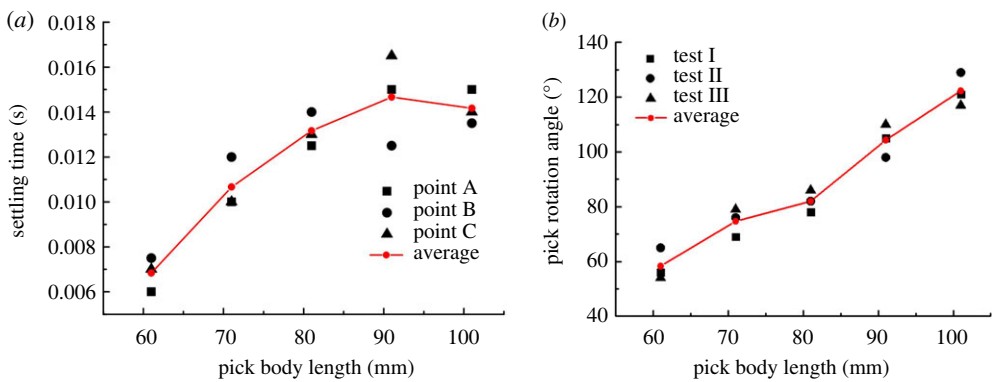

**Figure 10.** (a) ST and (b) rotation angle for different values of pick body length.

**Table 2.** Pick parameter values used to study effect of pick body length (units: mm).

| pick body length | pick handle length | pick handle diameter | inner diameter of pick holder |
|---|---|---|---|
| 61, 71, 81, 91, 101 | 62 | 34.67 | 35 |

width, the pick takes a long time to reach a stable state, and until then the pick collides with its holder and rebounds repeatedly. In this process, the frequency at which the cutting load acts on the pick changes a little and the load always decreases to zero, which results in CPOT changing a little.

Figure 9b shows that the trend of POTEP is mostly consistent with that of CPOT. This indicates that changing the gap width causes no change in the pick–holder interaction process (e.g. the change of contact forces), and the contacts between the pick and its holder at points A, B and C are similarly unaffected.

## 3.2. Effect of pick body length

Table 2 lists the values of the pick parameters used to study the effects of the pick body length.

### 3.2.1. Settling time

The $X$-direction displacement of point A and the $Y$-direction displacements of points B and C are assessed simultaneously, and displacement fluctuations of less than 0.02 mm are taken to indicate that the pick is becoming steady. Figure 10a shows how the ST of each of the three points on the pick varies with the pick body length.

Figure 10a shows that the average ST of the points on the pick increases initially with the pick body length and then falls slightly. The average ST increases gradually for pick body lengths of 61–91 mm and then falls slightly for the final pick body length of 101 mm. The increasing trend in the numerical results is consistent with that in the experimental results in figure 10b. The falling trend indicates that the other changes are due to changing the pick body length, which results in the nonlinear ST trend. The following analysis gives the detailed reasons.

### 3.2.2. Contact load

The contact load of the pick within 0.02 s was calculated using the contact forces of points A, B and C according to Eq. (7) in reference [37]. From the statistics, how the pick body length affects CPOT is shown in figure 11a, and how it affects POTEP is shown in figure 11b. From figure 11a, CPOT increases gradually with pick body length in different time periods. Except for the period of 0.005 s, CPOT tends to increase initially and then stabilize, which is consistent with the ST curve and agrees well with the experimental results (figure 10b showing how the pick body length affects the pick rotation angle). This verifies again the conclusion that the main factor in pick self-rotation is not the magnitude of the contact load but its duration.

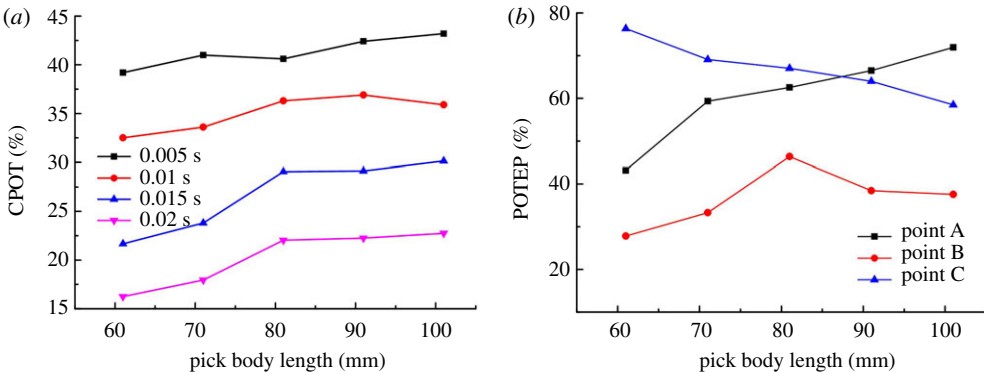

**Figure 11.** Effect of pick body length on (*a*) CPOT (*b*) POTEP.

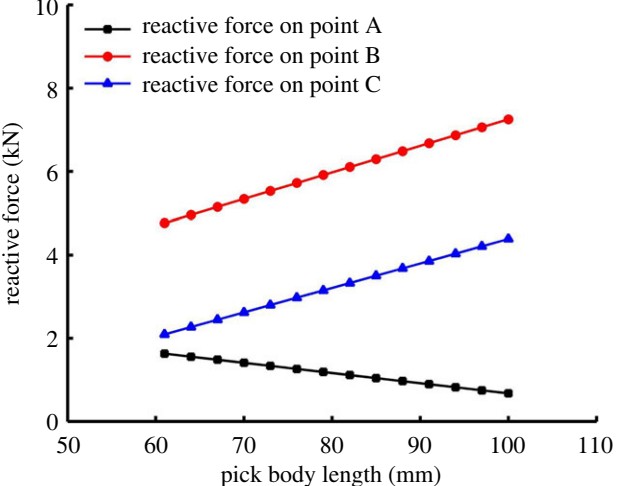

**Figure 12.** Effect of pick body length on reactive forces on points A, B and C.

It was pointed out above that the other changes were caused by changing the pick body length, which figure 11*b* verifies again. POTEP exhibits its own trends: at point A, it increases with the pick body length; at point B, it increases at first and then decreases; at point C, it exhibits a decreasing trend. The analysis shows that changing the pick body length causes changes in three aspects: (i) increasing the pick body length moves the centre of gravity of the pick towards the pick tip, which makes it easier for the pick to vibrate in its holder and enhances the pick's self-rotation ability; (ii) the inertia of the pick increases with the pick body length, which weakens both the vibration of the pick in its holder and the pick's self-rotation ability; and (iii) the reactive forces on points A, B and C are affected by the pick body length, as shown in figure 12 (Eq. (6) in reference [37]). The reactive force on point A exhibits a linearly decreasing trend, whereas the trends are increasing ones for points B and C. For small reactive force on point A, it can be concluded that the resistance of the pick moving in its holder increases, which also weakens the vibration of the pick in its holder and the pick's self-rotation ability. Furthermore, the ST and CPOT trends show that with increasing pick body length, the effect of the pick centre of gravity on the self-rotation ability is dominant in the early stage, and the pick inertia and resistance are the main factors in the later stage.

## 3.3. Effect of pick handle length

Table 3 lists the pick parameter values used to study the effects of the pick handle length.

### 3.3.1. Settling time

The *X*-direction displacement of point A and the *Y*-direction displacements of points B and C are assessed simultaneously. Figure 13*a* shows how the ST of each of the three points on the pick varies

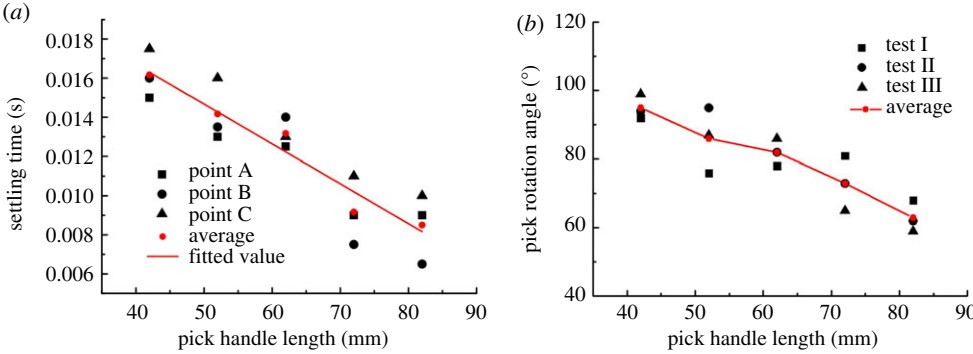

**Figure 13.** (*a*) ST and (*b*) rotation angle for different values of pick handle length.

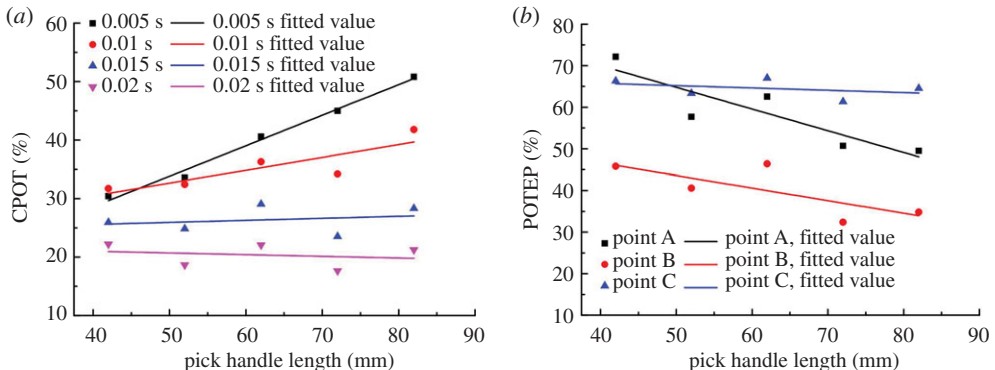

**Figure 14.** Effect of pick handle length on (*a*) CPOT and (*b*) POTEP.

**Table 3.** Pick parameter values used to study effect of pick handle length (units: mm).

| pick body length | pick handle length | pick handle diameter | inner diameter of pick holder |
|---|---|---|---|
| 81 | 42, 52, 62, 72, 82 | 34.67 | 35 |

with the pick handle length. The obtained data are so discrete that the ST lacks regularity. Thus, the ST trend is obtained by means of a linear fit to the discrete data. From the line in figure 13*a*, the ST decreases with the pick handle length, which agrees well with the experimental results in figure 13*b*.

### 3.3.2. Contact load

The effect of pick handle length on CPOT is shown in figure 14*a*, and the effect on POTEP is shown in figure 14*b*. From figure 14*a*, the trends of CPOT are different in different time periods. The results show an increasing trend with pick handle length in the periods of 0.005 and 0.01 s, almost no change in the period of 0.015 s, and a slightly decreasing trend in the whole period of 0.02 s. The latter trend is consistent with that of the experimental results (figure 13*b* showing how the pick handle length affects the pick rotation angle) and the results in figure 14*a*. The results indicate that the longer the pick handle, the more difficult it is for the pick to rotate. They also verify that the duration for which the pick experiences no contact load is the main factor in pick self-rotation.

Note that the trends in figure 14*a*,*b* were obtained by fitting the discrete data linearly. However, the tested data are largely fluctuant. It indicated the other changes were also caused by the change of the handle length. The analysis shows that changing the handle length similarly causes changes in three aspects: (i) the centre of gravity of the pick moves towards the pick root with increasing handle length, which makes it difficult for the pick to vibrate in its holder and weakens the pick's self-rotation ability; (ii) the inertia of the pick increases with the pick handle length, which weakens the vibration of the pick in its holder and the pick's self-rotation ability; and (iii) the reactive forces on points A, B and C are affected by the pick handle length, as shown in figure 15.

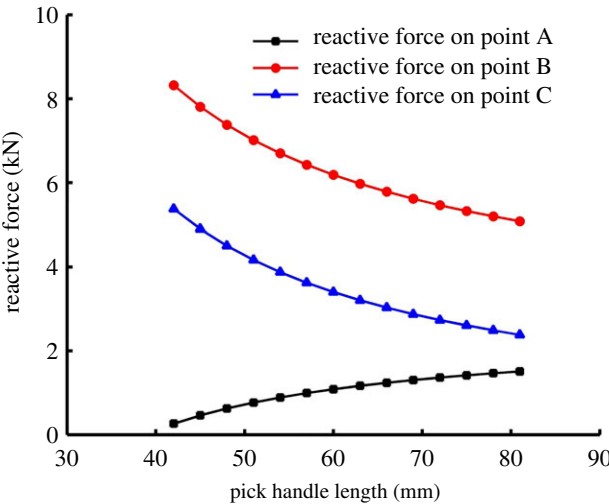

**Figure 15.** Effect of pick handle length on reactive forces on points A, B and C.

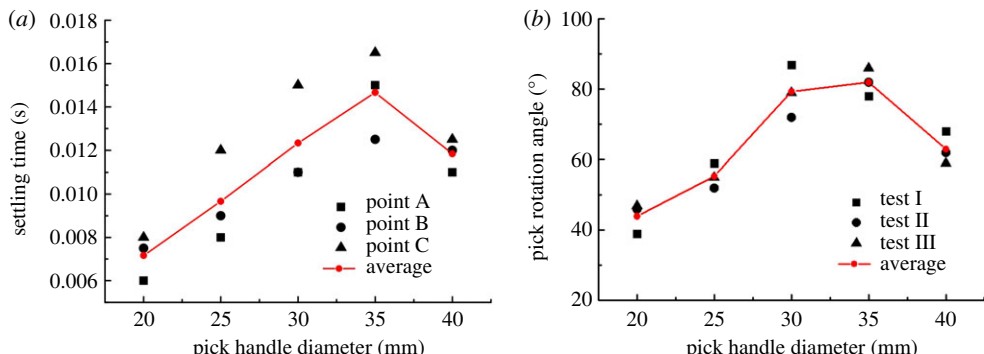

**Figure 16.** (*a*) ST and (*b*) rotation angle for different values of pick handle diameter.

**Table 4.** Pick parameter values used to study effect of pick handle diameter (units: mm).

| pick body length | pick handle length | pick handle diameter | inner diameter of pick holder |
|---|---|---|---|
| 81 | 62 | 19.67, 24.67, 29.67, 34.67, 39.67 | 35 |

The reactive force on point A increases with the pick handle length, whereas it decreases on points B and C. For small reactive force on point A, it can be concluded that the resistance of the pick moving in its holder decreases, which also increases the vibration of the pick in its holder and the pick's self-rotation ability. From the ST and CPOT trends, the effects of the pick centre of gravity and inertia on the self-rotation ability are dominant in the whole process of increasing the pick handle length.

## 3.4. Effect of pick handle diameter

Table 4 lists the pick parameter values used to study the effects of the pick handle diameter.

### 3.4.1. Settling time

The $X$-direction displacement of point A and the $Y$-direction displacements of points B and C are assessed simultaneously. Figure 16*a* shows how the ST of each of the three points on the pick varies with the pick handle diameter.

From figure 16*a*, the ST of each point on the pick increases initially and then decreases with the handle diameter. The ST increases gradually for diameters of 20–35 mm and then decreases sharply

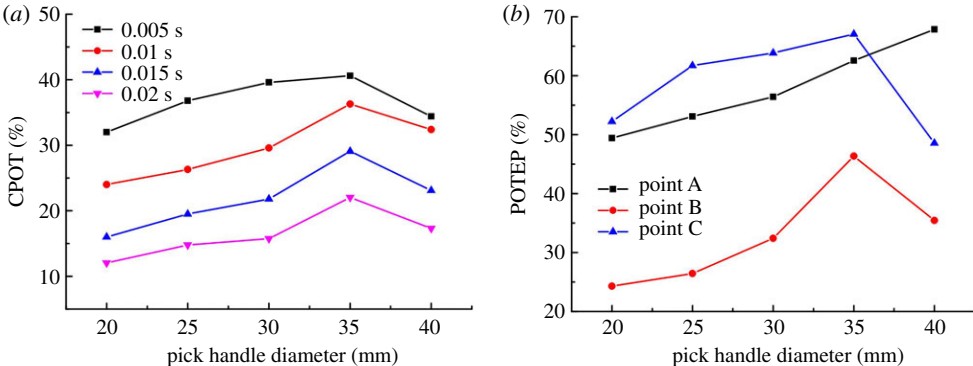

**Figure 17.** Effects of pick handle diameter on (*a*) CPOT and (*b*) POTEP.

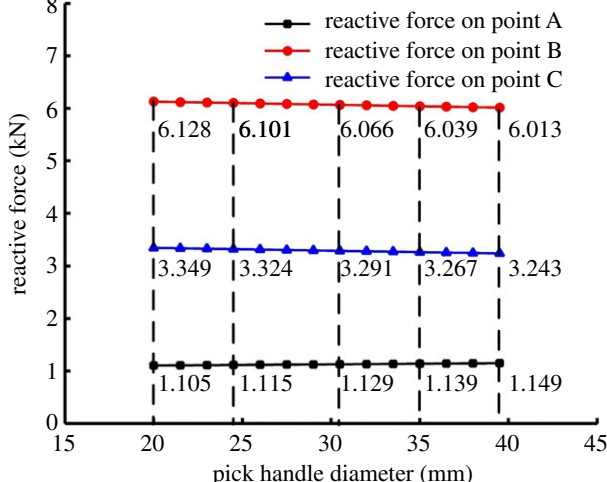

**Figure 18.** Effect of pick handle diameter on reactive forces on points A, B and C.

for the final diameter of 40 mm. The rising and falling trends in the numerical results are consistent with those in the experimental results in figure 16*b*. And it also indicated the other changes were caused by the change of the handle diameter, which made the trend of ST nonlinear.

### 3.4.2. Contact load

The effect of the handle diameter on CPOT is shown in figure 17*a*, and the effect on POTEP is shown in figure 17*b*. From figure 17*a*, CPOT in different time periods increases at first and then decreases with the pick handle diameter, which is consistent with the ST curve and agrees well with the experimental results (figure 16*b* showing how the pick handle diameter affects the pick rotation angle). Figure 17*b* shows that the other changes were caused by changing the pick handle diameter. POTEP exhibits its own trends: at point A, it increases with the handle diameter, whereas at points B and C it increases initially and then decreases.

According to the analysis, changing the pick handle diameter similarly causes changes in three aspects: (i) the centre of gravity of the pick moves towards the pick root with increasing pick handle diameter, which makes it difficult for the pick to vibrate in its holder and weakens the pick's self-rotation ability, but the pick handle diameter has less effect on the centre of gravity of the pick because of the pick's structure; (ii) the inertia of the pick increases with the pick handle diameter, which weakens the vibration of the pick in its holder and the pick self-rotation; and (iii) the reactive forces on points A, B and C are also affected by the pick handle diameter, as shown in figure 18.

The reactive force on point A increases with the pick handle diameter, whereas those on points B and C decrease. For small reactive force on point A, it can be concluded that the resistance of the pick moving in its holder decreases, which also increases the vibration of the pick in its holder and the pick self-rotation. From the ST and CPOT trends, with increasing pick body diameter, the effect of the resistance on the self-rotation ability is dominant in the early stage, and the centre of gravity and inertia of the pick are the main factors in the later stage.

# 4. Conclusion

(i) From comparing the ST and rotation angle under different parameters, it can be concluded that the duration for which the pick experiences no contact load is the main factor in pick self-rotation.

(ii) The ST of the pick increases with the width of the gap between the pick handle and its holder, and the self-rotation differs greatly within a gap width of 1 mm. With increasing gap width, all the CPOTs increase gradually in different time periods, and the trend of POTEP is mostly consistent with that of CPOT. This indicates that changing the gap width causes no change in the interaction process between the pick and its holder, such as changing the contact forces, and the contacts between the pick and its holder.

(iii) The ST of different points on the pick increases initially with the pick body length and then falls slightly. All the CPOTs increase gradually with pick body length in different time periods, while the POTEP has its own trends. This means that the other changes were caused by changing the pick body length. The ST and CPOT trends show that with increasing pick body length, the effect of the pick centre of gravity on the self-rotation ability is dominant in the early stage, and the inertia and resistance of the pick are the main factors in the later stage.

(iv) The ST decreases with the pick handle length. The trends of CPOT are different in different time periods, but the trend in the whole period of 0.02 s is consistent with that of the experimental results and the trend of POTEP. The results indicate that the longer the pick handle, the more difficult it is for the pick to rotate. Figure 14 indicates that the other changes were caused by changing the handle length. From the ST and CPOT trends, the effect of the pick centre of gravity and inertia on the self-rotation ability are dominant in the whole process of increasing the pick handle length.

(v) The ST of different points on the pick increases initially and then decreases with the handle diameter. All the CPOTs in different time periods increase initially and then decrease with the pick handle diameter, which is consistent with the ST curve and agrees well with the experimental results, while the POTEP has its own trends. This means that the other changes were caused by changing the pick handle diameter. From the ST and CPOT trends, with increasing pick handle diameter, the effect of the resistance on the self-rotation ability is dominant in the early stage, and the centre of gravity and inertia of the pick are the main factors in the later stage.

Data accessibility. The datasets supporting this article have been uploaded as part of the electronic supplementary material.

Authors' contributions. X.L. participated in the design of the study and drafted the manuscript. Q.G. participated in data analysis. All authors gave final approval for publication.

Competing interests. We have no competing interests.

Funding. This research is supported by the National Natural Science Foundation of China (51705029), the Fundamental Research Funds for the Central Universities, CHD (300102250203), the Natural Science Foundation of Shaanxi Province (2020JQ-375) and China Postdoctoral Science Foundation (2019M653515).

Acknowledgements. We thank Tang Ping (Lecturer at Xi'an Vocational University of Automobil) for editing the manuscript.

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
