## [Reviewer comments · Royal Society Open Science]

Review History

RSOS-200362.R0 (Original submission)

Review form: Reviewer 1

Is the manuscript scientifically sound in its present form?

Yes

Are the interpretations and conclusions justified by the results?

Yes

Is the language acceptable?

Yes

Do you have any ethical concerns with this paper?

Yes

Have you any concerns about statistical analyses in this paper?

No

Recommendation?

Accept as is

Comments to the Author(s)

I congratulate the authors. The manuscript is a result of high academic effort and contains valuable information for machine manufacturers and practicing engineers.

Review form: Reviewer 2

Is the manuscript scientifically sound in its present form?

Yes

Are the interpretations and conclusions justified by the results?

Yes

Is the language acceptable?

No

Do you have any ethical concerns with this paper?

No

Have you any concerns about statistical analyses in this paper?

No

Recommendation?

Major revision is needed (please make suggestions in comments)

Comments to the Author(s)

The core of the manuscript is scientifically sound by using the finite element method to study the effect that several technical specifications of conical picks and their respective holders have on the rotation of the picks while cutting rock. This is important for the efficient use of such cutting tools in terms of both technical and economical performance indicators because tool wear is dependent on pick rotation. Thus, the manuscript provides a useful contribution to the literature.

However, a thorough proofreading is required. The quality of the language makes it too difficult to understand the scientific content.

Some remarks:

1. In many occasions "cashed" is used instead of "crashed". Maybe "impact" or "collision" describes better the phenomenon. In many occasions "deceased" is used instead of "decreased".
2. The term "steady time" is confusing. One would expect this to refer to the "steady state" of the system. However, here it is used to describe the duration of transient phenomena before the steady state is reached.
3. The readability of Figure 18 is extremely low. To justify the relevant discussion either a different scale should be used for the reactive force axis, or the data should be tabulated.
4. The main finding is that contact duration has a stronger effect on rotation than the magnitude of contact load. In my opinion, the work is better described with the following title: "Effect of contact characteristics on the self-rotatory performance of conical picks based on impact dynamics modelling".

Decision letter (RSOS-200362.R0)

06-Apr-2020

Dear Dr Liu,

The editors assigned to your paper ("Effect of contacts Load on self-rotatory performance of conical pick based on impact dynamics model") have now received comments from reviewers. We would like you to revise your paper in accordance with the referee and Associate Editor suggestions which can be found below (not including confidential reports to the Editor). Please note this decision does not guarantee eventual acceptance.

Please submit a copy of your revised paper before 29-Apr-2020. Please note that the revision deadline will expire at 00.00am on this date. If we do not hear from you within this time then it will be assumed that the paper has been withdrawn. In exceptional circumstances, extensions may be possible if agreed with the Editorial Office in advance. We do not allow multiple rounds of revision so we urge you to make every effort to fully address all of the comments at this stage. If deemed necessary by the Editors, your manuscript will be sent back to one or more of the original reviewers for assessment. If the original reviewers are not available, we may invite new reviewers.

- Data accessibility

<http://datadryad.org/submit?journalID=RSOS&manu=RSOS-200362>

- **Competing interests**

- **Authors' contributions**

- **Acknowledgements**

- **Funding statement**

Kind regards,

on behalf of Professor Zach Agioutantis (Associate Editor) and R. Kerry Rowe (Subject Editor)
openscience@royalsociety.org

Editorial Comments to Author:

As you have been requested to edit the written English, you must provide proof that you have done so: acceptable proof includes a certificate of language-editing from a language editing service or a signed letter from a native speaker of English. If you do not provide this proof, your manuscript may be returned to you.

For information about language editing services endorsed by the Royal Society, please follow the link below:

<https://royalsociety.org/journals/authors/language-polishing/>

Associate Editor's comments (Professor Zach Agioutantis):

The paper needs to be proofread by a Professional Editor with technical knowledge in the subject area.

As it stands now, there are many typos and also some technical terms are not correct.

Reviewers' Comments to Author:

Reviewer: 1

Comments to the Author(s)

I congratulate the authors. The manuscript is a result of high academic effort and contains valuable information for machine manufacturers and practicing engineers.

Reviewer: 2

Comments to the Author(s)

The core of the manuscript is scientifically sound by using the finite element method to study the effect that several technical specifications of conical picks and their respective holders have on the rotation of the picks while cutting rock. This is important for the efficient use of such cutting tools in terms of both technical and economical performance indicators because tool wear is dependent on pick rotation. Thus, the manuscript provides a useful contribution to the literature.

However, a thorough proofreading is required. The quality of the language makes it too difficult to understand the scientific content.

Some remarks:

1. In many occasions "cashed" is used instead of "crashed". Maybe "impact" or "collision" describes better the phenomenon. In many occasions "deceased" is used instead of "decreased".
2. The term "steady time" is confusing. One would expect this to refer to the "steady state" of the system. However, here it is used to describe the duration of transient phenomena before the steady state is reached.
3. The readability of Figure 18 is extremely low. To justify the relevant discussion either a different scale should be used for the reactive force axis, or the data should be tabulated.
4. The main finding is that contact duration has a stronger effect on rotation than the magnitude of contact load. In my opinion, the work is better described with the following title: "Effect of contact characteristics on the self-rotatory performance of conical picks based on impact dynamics modelling".

Author's Response to Decision Letter for (RSOS-200362.R0)

See Appendix A.

Decision letter (RSOS-200362.R1)

Dear Dr Liu,

It is a pleasure to accept your manuscript entitled "Effect of contact characteristics on the self-rotation performance of conical picks based on impact dynamics modelling" in its current form for publication in Royal Society Open Science. The comments of the reviewer(s) who reviewed your manuscript are included at the foot of this letter.

on behalf of Professor Zach Agioutantis (Associate Editor) and R. Kerry Rowe (Subject Editor)
openscience@royalsociety.org

Appendix A

Revisions

Dear editor,

Thanks for the Editors' and the reviewer' comments on our paper. We have studied the comments carefully and tried our best to revise our manuscript according to the comments. The responses to the comments are as follows:

Editorial Comments to Author:

As you have been requested to edit the written English, you must provide proof that you have done so: acceptable proof includes a certificate of language-editing from a language editing service or a signed letter from a native speaker of English. If you do not provide this proof, your manuscript may be returned to you.

Our manuscript was edited by Charlesworth from 8-Apr-2020 to 15-Apr-2020. Now, Our English has been improved and the corresponding corrections for sentences or words have been conducted in the revised manuscript.

If it is still unsatisfactory, I am glad to improve it again.

EDITORIAL CERTIFICATE

This document certifies that the manuscript below was edited for correct English language usage, grammar, punctuation and spelling by qualified native English speaking editors at Charlesworth Author Services.

Paper Title:

Effect of contact characteristics on the self-rotatory performance of conical picks based on impact dynamics modelling

Author:

xiaohui Liu

Date certificate issued:

April 15, 2020

Associate Editor's comments (Professor Zach Agioutantis):

The paper needs to be proofread by a Professional Editor with technical knowledge in the subject area. As it stands now, there are many typos and also some technical terms are not correct.

As in the above case, our manuscript was edited by Charlesworth. If it is still unsatisfactory, I am glad to improve it again.

Reviewers' Comments to Author:

Reviewer: 1

Comments to the Author(s)

I congratulate the authors. The manuscript is a result of high academic effort and contains valuable information for machine manufacturers and practicing engineers.

Reviewer: 2

Comments to the Author(s)

The core of the manuscript is scientifically sound by using the finite element method to study the effect that several technical specifications of conical picks and their respective holders have on the rotation of the picks while cutting rock. This is important for the efficient use of such cutting tools in terms of both technical and economical performance indicators because tool wear is dependent on pick rotation. Thus, the manuscript provides a useful contribution to the literature.

However, a thorough proofreading is required. The quality of the language makes it too difficult to understand the scientific content.

As in the above case, our manuscript was edited by Charlesworth. If it is still unsatisfactory, I am glad to improve it again.

Some remarks:

1. In many occasions "cashed" is used instead of "crashed". Maybe "impact" or "collision" describes better the phenomenon. In many occasions "deceased" is used instead of "decreased".

The words ("cashed" , "deceased") have been corrected. "collide" is used instead of "crashed".

2. The term "steady time" is confusing. One would expect this to refer to the "steady state" of the system. However, here it is used to describe the duration of transient phenomena before the steady state is reached.

The term "settling time" has been used instead of "steady time" to describe the duration of transient phenomena before the steady state is reached.

3. The readability of Figure 18 is extremely low. To justify the relevant discussion either a different scale should be used for the reactive force axis, or the data should be tabulated.

The exact data has been added in Figure 18 according to the suggestion.

Figure 18. Effect of pick handle diameter on reactive forces on points A, B and C.

4. The main finding is that contact duration has a stronger effect on rotation than the magnitude of contact load. In my opinion, the work is better described with the following title: "Effect of contact characteristics on the self-rotatory performance of conical picks based on impact dynamics modelling".

The title has been changed according to the suggestion.

'Effect of contact characteristics on the self-rotation performance of conical picks based on impact dynamics modelling'